# Revisiting One of the Oldest Orphanages, Asylums, and Indigenous Residential Boarding Schools: The Thomas Indian School at Seneca Nation

**DOI:** 10.3390/ijerph21091120

**Published:** 2024-08-25

**Authors:** Hayden Haynes, Theresa McCarthy, Corinne Abrams, Melissa E. Lewis, Rodney C. Haring

**Affiliations:** 1Onöhsagwë:de’ Cultural Center, Salamanca, NY 14779, USA; hayden.haynes@sni.org; 2Indigenous Studies Department, State University of New York at Buffalo, Buffalo, NY 14260, USA; tm59@buffalo.edu; 3Department of Indigenous Cancer Health, Roswell Park Comprehensive Cancer Center, Buffalo, NY 14203, USA; corinne.abrams@roswellpark.org; 4School of Medicine, University of Missouri, Columbia, MO 65201, USA; lewismeli@health.missouri.edu

**Keywords:** Indian boarding school, Indian residential schools, missing and murdered Indigenous women and children, historical trauma, intergenerational trauma, health disparity, strength, resiliency, survivor, Native American, First Nations, Indigenous

## Abstract

For Indigenous populations, one of the most recognized acts of historical trauma has come from boarding schools. These institutions were established by federal and state governments to forcibly assimilate Indigenous children into foreign cultures through spiritual, physical, and sexual abuse and through the destruction of critical connections to land, family, and tribal community. This literature review focuses on the impact of one of the oldest orphanages, asylums, and Indigenous residential boarding schools in the United States. The paper shares perspectives on national and international parallels of residential schools, land, truth and reconciliation, social justice, and the reconnection of resiliency-based Indigenous Knowledge towards ancestral strength, reclamation, survivorship, and cultural continuance.

## 1. Introduction

For Indigenous Peoples in the United States (U.S.), often called Native Americans or North American “Indians”, and First Nations, Inuit, and Metis of Canada, the most recognized historical trauma came from residential boarding schools. The role of the boarding school was to advance the settler colonial project of acquiring Indigenous land for the economic, political, and settlement purposes of the U.S. This necessitated eliminating Indigenous life ways that were connected to and protected these lands. These processes and their consequences over time “induced or compelled” children to attend Indian Boarding Schools that were operated by religious institutions or organizations.

During the boarding school era, the main objective of the U.S. and Canadian governments can be summarized by a few statements written by government officials of the time:


*“A great general has said that the only good Indian is a dead one, and that high sanction of his destruction has been an enormous factor in promoting Indian massacres. In a sense, I agree with the sentiment, but only in this: that all the Indian there is in this race should be dead. Kill the Indian in him and save the man.”*


*–Colonel Richard Henry Pratt (USA), Founder, Carlisle Indian Industrial School, 1982* [1].


*“When the school is on the reserve the child lives with its parents, who are savages, and though he may learn to read and write, his habits and training mode of thought are Indian. He is simply a savage who can read and write. It has been strongly impressed upon myself, as head of the Department, that Indian children should be withdrawn as much as possible from the parental influence and the only way to do that would be to put them in central training industrial schools where they will acquire the habits and modes of thought of white men.”*


*–Canada’s Prime Minister, John A. MacDonald, 1883* [1].

Indigenous children were forcibly abducted, and, at other times, families surrendered children at the threat of legal action by the federal government. Similar boarding school assimilation practices were carried out by the church and each respective government authority against Indigenous populations across the U.S., Canada, Central and South America and the Caribbean, Australia, New Zealand, throughout Scandinavia, the Russian Federation, Asia, the Middle East, and Africa [2]. Globally, Indigenous children were taken from their homes, communities, and families and sent to institutions—some near and others far. The goal of the boarding schools was to “remove the Indian” and to replace Indigenous Knowledge with that of the colonizing society. In the U.S. and Canada, to accomplish this goal, children were not allowed to speak their Indigenous languages or practice Indigenous ways of life. They were forced to speak different languages, practice Christian religious behaviors, cut their long hair, and wear colonizer clothing. Choosing to disobey the rules or speak their Indigenous language resulted in consequences such as withholding food, solitary confinement, corporal punishment which at times led to severe psychological harm, physical injury, and/or death. Punishment was enacted in front of other students to frighten and intimidate them, as well as to subject recipients of punishment to public humiliation. At times, students were asked to carry out the whipping on their fellow peers, a practice that was not part of traditional parenting styles [3,4]. 

Indigenous boarding schools across the country were unhealthy places for students. In many, lives were lost due to the spread of communicable diseases (e.g., tuberculosis, trachoma, influenza), over-crowding, and poor nutrition (e.g., government issued food)—which, combined, contributed to depressed immune systems. For many schools, including Thomas Indian School (TIS), the poor health conditions were exacerbated by inadequate funds and insufficient trained health personnel [5,6]. During an interview, TIS student, Mary Pembleton (Tuscarora Nation), shared:

*“The oatmeal was wormy; the salt pork was cooked and served in its own grease; the beans and potatoes (with green spots) weren’t done. When they tried to mash the potatoes, it would shoot across the room, so the kids called them ’bullets’—steel, grey, and hard. They called the tapioca pudding ’frog eyes’. The bread was so hard that she could only eat the center, and if she wanted another slice, she had to finish the first slice completely. Mary’s baby teeth snapped off from biting into the hard crust. She ruined her teeth further by cracking prune pits with her teeth, just so she could eat the center of the pit. Mary’s teeth were bad by the time she left TIS.”* [7].

In 2021, the remains of 215 Indigenous children were allegedly discovered at the site of the former Kamloops Indian Residential School in British Columbia, Canada. Investigations like this are also underway in the U.S. [8]. The initial finding, using radar penetrating technology to locate this mass grave, which possibly holds the remains of children as young as three, serves as an agonizing reminder of the cruelty that Canadian and U.S. Indian policies enacted upon the youngest and most vulnerable citizens of Indigenous Nations. From the early 19th to the late 20th centuries, more than 350 government-funded and church-led Indian boarding schools operated in the U.S., while 139 such schools operated in Canada. As places of profound dehumanization, with mortality rates reaching as high as 60%, these schools and their infamy have inflicted untold suffering on Indigenous families, communities, and nations. The potential findings at Kamloops have ignited a wave of grief across Indigenous North America.

Childhood trauma is one of the clearest predictors of health and well-being in later life [9,10,11,12]. Indigenous childhood experiences in boarding schools contributed to the poor health and well-being of those attendees into adulthood and for Indigenous people today. Boarding school attendance is related to increased thoughts and experiences of historical loss and trauma, which are related to emotional health concerns including depression, suicidal ideation, and intimate partner violence [13,14,15]. The trauma experience from boarding schools has been found to be passed down intergenerationally, and even increase for each subsequent generation, revealing the continued and persistent impact of these events on Indigenous children today [11,15,16,17,18,19]. 

In addition to emotional damage, the experience of boarding school attendance is related to increased physical health disparities compared to those who did not attend boarding school, including pain, physical limitations, and functioning, as well as overall poor health [20,21]. (Further, those who attended residential boarding schools as youth were three times more likely to have cancer as adults [22]. Another example of historical trauma resulting from boarding school experiences is the negative impact on trust and limited utilization of healthcare resources. To address cancer health and outcome disparities, it is clear that trust must be rebuilt, as must the lacking infrastructure, allowing Indigenous Peoples to access high-quality cancer care [23].

While the boarding school era ended around 1950, the racist beliefs that underlie those years persisted in the eras which followed. The Indian Removal/Adoption era occurred between 1950 and 1980 when Indigenous children were displaced from their homes at a higher rate than any other racial/ethnic group, resulting in approximately 1/3 of all American Indian children being removed from their home. Social workers removed children due to the biased beliefs that American Indian people, communities, and schools were not safe places for children. For instance, children were removed from their homes for residing on a reservation, attending a tribal school, or if their home was multigenerational [24]. Even today, Indigenous children are placed in foster care at a 2.6 times higher rate than the general population and are 4 times more likely to be placed in the welfare system compared to White children [25]. Indigenous-specific stressors such as childhood trauma relate to the increased health risks today, from higher rates of depression to cardiometabolic disease [26,27]. 

Despite these cultural genocide attempts, such traumatic experiences also highlight the strength of Indigenous Peoples and bolstered ingenuity, resiliency, and survivorship [3,4,25,28,29,30,31]. In fact, efforts are being made to address the field of social work to reduce the bias that contributes to their perpetuation of the intergenerational trauma of Indigenous children and allow Indigenous people, communities, and families to live according to their tribal lifeways.

It is important to provide a historical timeline of the origin and function of the boarding school-era policies. Colonization of the Americas was predicated on the false belief that Black people, Indigenous people, or People of Color were non- or sub-human [32]. The American Anthropological Association further shared that racism was invented in the Americas with the colonization of Indigenous Peoples and is a social construct to fulfill the needs of the colonizer [33]. This belief allowed for violations of international laws and ethics such as colonization, enslavement, and theft of land and property [34]. Throughout the colonial timeline of the U.S., different political eras (e.g., boarding school, Termination, Indian Adoption) continue to be inflicted on Indigenous Peoples with the same basic philosophy of white supremacy and racism—with the same goals of cultural genocide and assimilation.

This historical perspective paper focuses on the impact of one of the oldest orphanages, asylums, and Indigenous residential boarding schools, located on a Seneca Nation of Indians Territory (Reservation) adjacent to New York State. Materials guiding the approach and methods included a tribal workgroup that convened on a regular basis to discuss the orphanage, asylum, and boarding school. The group coalesced and shared primary and secondary documents, sources, texts, and papers. These combined documents were used as reference materials for the historical perspective.

### 1.1. Haudenosaunee Nations

The original Haudenosaunee Nations (Iroquois Confederacy) included the Mohawk, Oneida, Onondaga, Cayuga, Seneca, and later the Tuscarora. The Mohawks are known as the “Keepers of the Eastern Door” and are responsible for protecting and defending the eastern boundaries of the Haudenosaunee Territories. The Onondaga are the “Keepers of the Central Fire” and are the capital of the Confederacy. The Senecas are the “Keepers of the Western Door” and are responsible for protecting and defending the western boundaries [4,35]. These Indigenous Nations are bound together by the Great Law of Peace, which shares the collective effort and process of the Good Mind and peaceful action towards sustainable strength [36,37,38,39].

The Haudenosaunee are ancestrally, culturally, linguistically, and politically related and span across present-day New York, Pennsylvania, Ontario, and Quebec, Canada. The Haudenosaunee were the first Indigenous government to hold U.S. treaties, noted as the Supreme Law of the Land, with President George Washington and were also known to have influenced a significant part of the development of the new U.S. government, based on Haudenosaunee governance [40]. Although the Haudenosaunee have lost much of their land-base and were relocated, for the most part, they remain on their ancestral lands, which later changed title from Reservation to Territory status in the U.S.

The Haudenosaunee, specifically the Seneca Nation and the Six Nations of the Grand River (Ontario), were two of the first to encounter the U.S. and Canadian processes of assimilation by means of missionary, state, and federally driven orphanages, asylums, and boarding schools. These included the Thomas Indian School (TIS), located on the Cattaraugus Territory (Reservation) of the Seneca Nation and the Mohawk Institute in Brantford, Ontario, Canada. These institutions were initiated throughout the 1800s, coinciding with other Indian boarding school years, including the Carlisle Indian School (Carlisle, PA, USA) built for Indigenous children from across North America.

### 1.2. Missionaries, Orphanages, Asylums, and New York State’s Thomas Indian School at Seneca Nation (1855–1957)

As tribes, bands, and Indigenous Nations document more forensic evidence to confirm and support the radar penetration findings, initial reports of potential mass graves of many Indigenous Nations across North America initiated awareness campaigns regarding residential boarding schools, resulting in the 2021 establishment of Canada’s National Day for Truth and Reconciliation as “Orange Shirt Day” to promote awareness of and education on residential school systems in North America [41]. For the citizens of the Seneca Nation and other Haudenosaunee members, remnants of the Asylum for Orphan and Destitute Children (later renamed the Thomas Indian School) continue in the few living survivors who attended the school. Trauma inherited by the children and grandchildren of the survivors, as well as the buildings and locations on the Cattaraugus Territory, not only tell the story but also still haunt us today.

The painful history reminds us of the cultural genocide that took place and continues today. The history of Euro-American encroachment and the occupation of Haudenosaunee lands provides a contextual framework for the TIS era, although there is only space herein to engage this history in broad strokes. Seneca historian, Dr. Marilyn Schindler, characterizes European presence in Haudenosaunee lands as “pierc[ing] all Territories like a relentless thunderstorm”. This invasion of Turtle Island was extensive, she recounts, “So many millions died, and so much land was taken” [42]. For the Senecas, like other Indigenous Nations, this destruction of land and Indigenous ways of life was unremittent.

From their earliest encounters with European newcomers, the Seneca were forced to defend their relationship to their lands, claimed by rights of the European “Doctrine of Discovery”, a doctrine upheld by Christianity and enforced by violence. Originating from a series of 15th century Papal Bulls, the Doctrine was a construct of international law dictating that all lands uninhabited by Christians could be claimed by Christian monarchs. This formed the basis for colonial conquest. Today, the United States continues to uphold the Doctrine of Discovery, privileging settler rights to Indigenous lands though its judicial and legal systems. In recent decades, the provisions of the Doctrine were used to dismiss Cayuga, Oneida, and Onondaga land claims and actions in their ancestral territories. This legal fiction, and others that build upon it, remain an apparatus of U.S. federal law to this day [43,44,45]. Throughout history, Christianity never lost its association with the Doctrine of Discovery as it is administered to Indigenous Peoples. The church claimed to be a benevolent institution in their conversion of Indigenous Peoples to Christianity through structures like the TIS. These harmful and deadly ideologies advanced the settler interests in Seneca Lands. Accordingly, Seneca homelands were under a constant threat of invasion long before the emergence of Indian boarding schools.

For the Seneca, the establishment of the schools occurred after “centuries of disruption, war, fraud, and deception” [46]. As the late Professor John Mohawk (Seneca Nation) wrote in his book, War Against the Seneca: The French Expedition of 1687, the Haudenosaunee emerged successful in maintaining control of their territories against New France, although French soldiers laid waste to Seneca Country [47]. Recounting the Seneca experience of the 1779 Clinton–Sullivan campaign, Seneca historian and artist, G. Peter Jemison, quotes George Washington’s order directly, as he demanded the “total destruction and devastation of their (Seneca) settlements and the capture of as many prisoners of every sex and age possible” [48]. With the destruction of forty villages and the decimation of their crops, the Seneca and other Haudenosaunee who managed to escape sought refuge at Fort Niagara and Buffalo Creek. In the aftermath of the American Revolution, the Seneca were coerced into large-scale land surrenders without remuneration [49,50]. Their land rights and interests were completely ignored in treaties among colonial powers, like the Treaty of Paris (1783) [51]. Throughout the eighteenth and nineteenth centuries, Seneca Lands continued to be targeted by land speculators seeking profit from their sale to the region’s growing settler population [46]. It is within this context that the TIS was crafted and constructed and, for the purpose of this paper, serves as a historical case study.

The influences of Reverend Asher Wright figure prominently in many historical accounts of the era of the TIS’s establishment. Wright, a Presbyterian missionary, worked among the Seneca Nation, initially in the Buffalo Creek region, which accounted for most of the Genesee Valley, NY, to the present-day area of Western New York State [52]. In 1832, Wright advocated for the Seneca during negotiations for the Buffalo Creek Treaty, noting the treaty was corrupt due to land developers bribing Seneca leaders [53]. However, Wright’s translations and suggestions for a new Seneca Constitution were strongly opposed by the Seneca hereditary Council of Chiefs (Allegany/Cattaraugus), who shared their concern regarding Wright’s missionary influence changing the political structures of the Seneca. Further, the Chiefs opposed Wright’s interference in their political matters, particularly his suggestions for a new Seneca Constitution [54].

According to Fenton and other anthropologists, this conflict led to the dissolution of government by life chiefs and the establishment of a republic with laws and a constitution called the Seneca Nation, incorporated under the State of New York, which featured a majority rule which replaced the older system of unanimity. Robert’s Rule of Parliamentary Procedure thus became the order of the Seneca Council for the Allegany and Cattaraugus Reservations (Territories) [52]. A less simplistic and more nuanced interpretation of these times would also account for the difficult and tumultuous history of violence the Seneca endured. It would foreground Seneca experiences of invasion, land theft, forced relocation, militarized brutality, and the cohesion, duplicity, and lies that marked all the diplomatic engagements with Euro-American settlers. These contexts are crucial for understanding colonial entanglements, and Seneca efforts to navigate colonial structures to meet the needs of the people.

The 1838 Treaty of Buffalo Creek dissolved the Buffalo Creek Reservation and forced residents to relocate [55,56]. By 1845, the Buffalo Creek Seneca had relocated, with many going to the Cattaraugus Territory and some to Allegany—both locations became part of the Compromise Treaty of the current day Seneca Nation of Indians. It was also during this era (1797–1836) that the English language started to become commonly used among the Seneca [3,52].

In 1854, a Seneca man passed away leaving ten orphaned children and it was learned there were forty more orphans among the Seneca communities. Rev. Wright and his wife Laura (also a missionary and founder of the Iroquois Temperance League), in response to this need, founded the Asylum for Orphan and Destitute Children in the summer of 1854. This was later to be chartered by the state in 1855 on the Cattaraugus Territory and is likely the first New York State-chartered entity on sovereign Seneca lands [3,55]. “Within a few months after the completion of the asylum, a resolution was passed requiring that the English language be used exclusively” [57].

The Asylum for Orphan and Destitute Children at Seneca on the Cattaraugus Territory predates the Carlisle Indian School, PA (1879), by nearly a quarter-century. As the school grew, Phillip E. Thomas, Quaker Society of Friends Chairman of Indian Affairs, first president of the Baltimore and Ohio Railroad and the third son of Evan and Rachel (Hopkins) Thomas (relatives of the founder of Johns Hopkins University), provided financial support. These efforts and contributions led to the asylum being renamed the TIS. Rev. Wright was director until 1875, when it passed to the New York State Board of Charities [3,6,52]. This board was tasked with the assimilation, education, and vocational training of Indigenous children in its care. This also meant that it had custodial care of the children, meaning that for children to attend, their parents had to sign over guardianship. Attendees of the asylum were referred to as inmates.

By 1871, TIS had 80 students, and 104 by 1895. In 1905, eight grades were offered at the school, which had a half-day system with children attending classes for half the day and working as child labor for the school’s facilities for the other half. By 1915, 213 students attended and in 1930, the boarding and residential institution was classified as a junior high school. Within the first 40 years, 951 students passed through its doors. In total, 2500 students were received and discharged between 1855 and 1955 [58]. TIS was eventually closed in 1957 by New York State [6,59]. Nearly all the Indigenous children who attended the school were from the Haudenosaunee [55]. 

The colonizing military fashion of instruction paired with physical, mental, spiritual, and emotional maltreatment was exceptionally damaging to boarding school residents and day-schoolers. Students were subject to rigid time schedules and whistles were used to signal when it was time to move. One TIS attendee shared the school was “a place where no one loved you” [55]. Residential students were pitted against day students, encouraging lateral individual, family, and community violence (Marguerite Lee-Haring, Seneca Nation of Indians, about experiences as a day student at the TIS [60]).

The process of Christianizing Indigenous youth forcibly took away Indigenous language, religion, cultural, land-based learning, traditional diets, wellness practices, and spiritual beliefs. Many children ran away, only to be found and returned due to the often-ill-obtained custodial rights of New York State. After the school closed in 1957, the Gowanda New York State Hospital used the reservation-territory-based TIS buildings for non-Indigenous psychiatric patients. Today, of the dozens of buildings that once stood within the 300-plus acre area, only a few buildings remain, but the effects of the school continue to be felt to this day [57].

### 1.3. Childrens’ Schools: Cemeteries

Recent news of the alleged remains of children at Kamloops drew international attention to a devasting and long over-looked reality at the residential school for Indigenous children, which had cemeteries while the schools of their white counterparts had playgrounds. The alleged findings connected to the radar-penetrating discovery of deceased boarding school children disposed of in unmarked graves are a reminder of that unresolved trauma. Indigenous People are still grieving the loss of so many of our children. The TIS campus has maintained ties with the adjacent church and Tribal government burial grounds.

In an era of Truth and Reconciliation, and the Missing and Murdered Indigenous Women and Children (MMIW) movement, Indigenous Nations in Canada have unearthed the historical truths of cemeteries in, at, or adjacent to federal, state, provincial, and church-driven residential schools. Truth and Reconciliation is needed in the U.S., as referenced in the Federal Indian Boarding School Initiative Investigative Report [30]. At Seneca, these boarding schools include burial grounds at Cattaraugus, Seneca Nation, including in the church cemetery. These burial plots are in or near the former TIS recess areas [59]. Other cemeteries at Seneca tied to the boarding school era include church cemeteries, whose graveyards extend into the dense old forest, with unmarked plots and markers of Indigenous warriors/veterans from the Grand Army of the Republic—the Union military who served in the American Civil War. The unmarked graves in these plots potentially tied to the 1847 typhoid outbreak [55].

### 1.4. The Role of Social Justice in Achieving Indigenous Equity/Sovereignty

Revisiting historical perspectives through social sciences with shifts towards decolonization and reconnection with Indigenous Knowledge are important within the framework of social justice, equity, and sovereignty. For example, the rate of Indigenous child out-of-home placement in the 1970s was 25 times higher than the rate for non-Indigenous children and most placements (90%) were to non-Indigenous homes [24]. Indigenous people viewed the field of social work as complicit in these practices to inequitably remove Indigenous children from their families and their culture and both challenged and entered the field to protect Indigenous communities. Indigenous social workers called attention to the high rate (1/3) of Indigenous child home displacement by social services between the 1950s and 1970s and called for federal intervention, which is now known as the Indian Child Welfare Act [61]. This Act was passed in 1978 to exercise tribal sovereignty over non-Indigenous systems’ decisions around child out-of-home placement.

At present, the preamble for the National Association of Social Work (NASW), *Code of Ethics*, states that the mission of the field of social work is to enhance human well-being and the empowerment of people with a duality of focus on individual well-being and the well-being of society [62,63]. As such, social workers bi-directly interact with societies to promote and understand social justice and change on behalf of and with people and communities. These pathways include forms of culturally aware advocacy, social action, policy development, education, and research.

Cultural genocidal practices continue today and can still be seen in policies and practices affecting Indigenous children. Justice work is desperately needed today with the inequitable rates of the out-of-home placement of Indigenous children and violations of the Indian Child Welfare Act. Further, Osage scholar Alex Redcorn explains that boarding schools of the past are not needed anymore because every school in the United Stated is already functioning as a boarding school, conveying the fact that most Indigenous children (93%) attend public schools in the United States with practically no education on Indigenous history, culture, or language, resulting in the same experiences as Indigenous children in past boarding schools: cultural shame, humiliation, isolation, and sub-par educational experiences resulting in poor academic achievement. This all occurs despite the knowledge that Indigenous children fare better when education is culturally tailored. Without clear and immediate reform to policies and practices, Indigenous children will continue to experience oppression, separation from their family and tribe, sub-standard education, healthcare disparities, and a continuation of the trajectory of cultural genocidal policies of the past [64,65,66].

One core value of the profession is social justice. NASW shares this as an ethical principle to challenge social injustice or in an Indigenous framework to encourage resiliencies. In pursuit of social change, efforts are being made to promote knowledge about oppression (e.g., boarding schools, Indigenous Knowledge-based research) and increase diversity knowledge on all planes. The goal is to ignite a change from social inaction to culturally rallied social action in a respectful yet persistent manner. This can be in the form of access to services, resources, equality (in research process), meaningful participation, and decision-making for the individual, family, clan, community, or Indigenous Nation.

In an Indigenous framework, ethics and mindset can be framed by the ancestral knowledge housed in Indigenous principles. One example is the Haudenosaunee principle of the “Good Mind” towards peaceful action to encourage strength, power, and righteousness. Combined, these actions create a balanced approach for finding resiliencies to improve social welfare and increase cultural and intellectual prosperity for future generations. A reflection of this is shared by Indigenous leader, Michael Martin [67].

In line with the UN Declaration of Indigenous Human Rights, which points out the imperative importance of Self-Determination, the NASW echoes this as a core responsibility in social justice, equity, and the ethical responsibility to respect and promote the right of a community to self-determination and assist in a global effort to identify goals to achieve this [62]. Further, social action is recommended in the form of sovereign governance and engagement in political advocacy for equal access to basic human needs.

Lastly, it is not only the social worker’s’ ethical responsibility to the broader society, but all peoples’ responsibility—past and present—to promote the general welfare of society, from local and tribal to global levels, for the continued growth of humankind, collective healthy mind sets, and our environments [62]. In the words of Linda T. Smith (Māori), “Supporting the development and engagement of new Indigenous warriors can promote the processes of transformation, decolonization, healing and mobilization for social justice” [68].

### 1.5. Good Mind: Peace, Power, Strength, Resilience

Since the beginning of time, the Haudenosaunee have an inherent process of healthy mind-setting and re-connection to resiliency [69]. Prior to the process of attempted colonization, genocide, and boarding schools, the Haudenosaunee embraced ideas of the Great Law of Peace along with strength-based teachings that surrounded collective thought toward community togetherness for peace, strength, protection, and harmony [70,71]. To the Haudenosaunee, peace is more than just the absence of conflict but is founded on spiritual and social foundations of wellness and is active work toward establishing universal justice [72]. This ancient process of Indigenous theory and practice was and is a powerful tool in the survivorship and resilience of Haudenosaunee children, now adults, who attended the asylums, orphanages, and boarding schools and provides a continued pathway towards a reconnection to Indigenous Knowledge and space [73]. In summary, residential and boarding school survivors of today are the foundations of *Yonkwa’nikonhrakontáhkwen* (Continuance), “Our consciousness continues unchanged”, portraying cultural strengths and meaningful ways to live in the world with balanced harmony, embodying continuance in the health of the mind, body, and spirit [74,75].

## 2. Discussion

The Seneca, of the Haudenosaunee Confederacy, have faced a long trail of war, forced removal, displacement, and attempts at forced assimilation and genocide. In Seneca Country, the first removal included the U.S. military’s destruction and removal of Genesee Valley homelands by George Washington, known to the Haudenosaunee as “Town Destroyer”; a second relocation to Buffalo Creek Treaty Lands; a third forced relocation to the Cattaraugus and Allegany Territories (Ohio Valley and later to Wisconsin, Kansas and Oklahoma); and a fourth relocation for some families during the 1964 Kinzua Dam removal from Allegany Seneca [76,77,78,79,80]. From the Sullivan–Clinton Campaign (1779) to the current day, the Seneca People have faced the generational displacement of land-base, language, ancestral diet, and spirituality, and ancestral burial ground displacement—which all contribute to current day historical traumas, resulting in trans-generational effects. These translate to epi-genetic connections related to health disparities but also strength, resiliencies, and protective factors [48,81].

International and national organizations that work to restore Indigenous rights include the United Nations, the National Congress of American Indians (NCAI), and the Native American Right Fund (NARF) [82,83,84]. The top priorities for NARF include preserving Tribal existence, including advocating for the basic human rights of Indigenous People and holding governments accountable to Indigenous Nation laws and treaties. The United Nations Declaration on the Rights of Indigenous Peoples (UNDRIP), adopted in 2007, is the most comprehensive international instrument on the rights of Indigenous Peoples. It establishes a universal framework of minimum standards for the survival, dignity, and well-being of the Indigenous Peoples of the world and it elaborates on existing human rights standards and fundamental freedoms as they apply to the specific situation of Indigenous Peoples [82].

Containing language specific to residential boarding schools, UNDRIP states in Article 3 of the declaration that Indigenous Peoples have the right to Self-Determination. By virtue of that right, we freely determine our political status and freely pursue economic, social, and cultural development. Further, Article 7, Section 2, shares that Indigenous Peoples have the collective right to live in freedom, peace, and security as distinct peoples and shall not be subjected to any act of genocide or any other act of violence, including forcibly removing the children of the group to another group, and Indigenous Peoples and individuals have the right not to be subjected to the forced assimilation or destruction of their culture (Article 8) [82]. Combined, these UN statements support the idea of truth, reconciliation, and social justice.

Indigenous Peoples were not and often are not permitted to be experts on Indigenous lives, experiences, and research and over time that authority was thought to be held only by non-Indigenous Peoples. It is a clear function of the mainstream view of Indigenous People as less worthy, valued, and human that these stories were not heard before the physical evidence was unearthed, and the reality could no longer be denied. This reality is no longer vague to the public, while it never was to Indigenous People who spoke of it through narratives of their own experiences or even the silences in families about these experiences, which also spoke volumes. It is truly time to rethink the qualification of “cultural” genocide and change it to stand as “genocide” or include physical genocide in the term. Genocide is a form of mass-organized violence and confronting the truth means not sanitizing this or limiting its scope. The next steps include (1) a thorough inquiry into settler-colonial actions; (2) a tribally led forensic investigation; and (3) increased support, efforts, and resources towards Indigenous healing.

## 3. Conclusions

The Haudenosaunee have always held the principles of community wellness and caretaking. An anonymous writer cited in Van Meteren (1535–1612) wrote the following personal account of meeting with the Haudenosaunee:

“We went another league and a half and came to a hunter’s cabin, which we entered to eat some venison a Chief invited us into his castle. There was a big fire lighted, and a fat haunch of venison cooked, of which we ate. He gave us two bearskins to sleep upon and presented me with three beaver skins. We slept in this house, ate heartily of pumpkins, beans and venison, so that we were not hungry, but were treated as well as possible in their land” [85].

This passage presents a historical foundation for the Haudenosaunee’s social responsibility in the form of food sharing; in this case, across racial boundaries [73]. It is time to re-think and analyze the ethical and moral behavior of settler-colonists and Indigenous People to reflect the truths of history and today versus false narratives (e.g., savages) created out of greed. Although external influences and the politics of the residential boarding school era and multiple removals of the Seneca may have been a chapter in the narrative of Haudenosaunee and American/Canadian history, it is but only one chapter in the existence and continuance of a society.

As future chapters emerge and are co-created with historical context, traditional perspectives are braided into Indigenous Knowledge and Western teachings. The practice of and re-connection with Indigenous Knowledge is one means of addressing systematic colonization and the boarding school legacy. Many Indigenous communities are skillfully adapting to create new narratives that inspire strength, hope, and love, narratives that help individuals, families, and communities to “find their medicine” in reconnecting with their ancestral lifeways. In Seneca and other Haudenosaunee societies, this translates to the re-birth, continuance, and re-connection to land-based learning in the forms of planting, lacrosse stick making, subsistence fishing, hunting, berry-picking, and medicine collection. It is also represented by finding resiliency in reconnecting with ancestral ways of living through foods, dance, game, song, and language. While this is clear to Indigenous community members, Western research has also demonstrated that a return to Haudenosaunee lifeways, such as food systems, results in improved physical health, mental health, and community well-being [86].

Among the Haudenosaunee and many other Indigenous Nations, we not only honor our past relatives, children, families, and societies, as well as our current generation, but embrace survivorship, continuance, strength, peace, and appreciation for future generations of people, life, and the symbiotic relationships and parallels with our natural world, as noted in the Haudenosaunee Ganon:yok or “Thanksgiving Address” that offers gratitude from the earth to the celestial realms. The Haudenosaunee have given many gifts to the world (e.g., Thanksgiving concepts, constitutions, principles of peace) but have so much more to offer that could heal people, water, and lands alike.

Ancestral knowledge and memory have the ability to recognize and remember the harm and carry intergenerational traumas of the past but, more importantly, to remember the strength, resistance, resilience, and the love of our ancestors from the beginning of time and continuing through the period of enforced placement in institutions like the TIS. A walk through the forest reminds us of this each season through the awakening of spring, exemplified by the berries, some carrying on the same branch a fruit of medicinal healing surrounded by the thorns of caution. Teachings about animals, birds, and humans alike, show that both good and clouded lives can co-exist in the same environment, and that people and populations must be cognizant of the harms and traumas but also of the medicine of strength and hope.

It is these understandings, remembrances, connections to the land, and singing of ancient songs that have survived from time immemorial that bring the melodies of peace, harmony, and survival that bridge and highlight the importance of ongoing truths. The honoring of residential school survivor communities and of the last resting places of the children of residential schools is part of our present-day responsibility and an integral part of Haudenosaunee, Indigenous Nations, American, Canadian, and global history.

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
