# Peer review of "Revisiting One of the Oldest Orphanages, Asylums, and Indigenous Residential Boarding Schools: The Thomas Indian School at Seneca Nation"

_ijerph, 2024, doi:10.3390/ijerph21091120_

Round 1
Reviewer 1 Report
Comments and Suggestions for Authors
Thank you for the opportunity to review this paper on a very important topic. This paper adds significantly to the growing literature on truth telling, making history visible and available internationally. I thank the authors on presenting this material, and can only imagine the personal journey to read all of the material and present it in this way.
I have given a lot of thought on how best I can respectfully give feedback on the structure of the paper, rather than the content, which is obviously very well and carefully researched.
Abstract
While the paper is a literature review it may be better to instead position it as a historical perspectives piece. (Abstract line 15)
Or, if preferring to present as a literature review, then details of the literature review search could be added. I personally perceive that it reads more powerfully as a truth telling historical perspectives piece, as is.
Introduction
Many First Nations and non-First Nations scholars internationally are now writing a standpoint at the beginning of paper, to clearly identify themselves and their standpoint. This would also strengthen this paper I believe. It is not until line 446 that it is is indicated that the authors are Haudenosaunee and/ or from other Indigenous Nations
Perhaps reword the first line to read For Indigenous people in the United States; often called Native Americans, Northern American … - is there a term that is preferred by First Nations peoples themselves?
The two quotes on page 1 certainly set the tone for the paper and strongly emphasise truth telling.
Line 45- was there also a situation where families took children to seek help (for serve illness) believing that the children would be returned? (There may or may not be, but has occurred in other countries as part of colonisation and child removal)
Line 47 – ‘to among’ - suggest remove among
Line 57 In America and
Line 60 …as well as subject [them] to public humiliation.
Line 62- perhaps the emphasis on the introduction of violent parenting styles could be emphasised a bit more – I am reading between the lines that the author's emphasis is that this is not the traditional First Nations approaches to parenting – this might be emphasised more strongly.
Line 72 – for an international audience, can you explain a little what is meant by government issued commodities.
Line 84- 86 – can you emphasis a bit more about how these experiences have led to people having cancer as adults.
Comment. You have described the terrible findings at Kamloops and wider implications well for an international audience. It rocked us all, and our thoughts are with First Nations peoples there in Canada, the US and worldwide.
Line 116 – 118 – as per previous comment, I suggest calling this a historical perspectives paper, which is powerful in itself.
Social justice line 298 – the paper changes direction here to more recent times and a focus on social work. I am wondering if a different sub heading may help indicate this shift. I perceive that this may have originally been a thesis chapter, and it would work for that, but for this paper it would help to indicate the change in focus differently.
Is this section about an Indigenous approach to social work and social justice?
Discussion
Having no information about the cultural back ground of the authors makes reading the discussion a little challenging. I ask myself, are the authors talking of self and ancestors, or of ‘others’.
Similar to the discussion line 402 – 404. It may be that the authors have clearly identified themselves, but that is blinded to the reviewers (which happened for us with a paper recently).
There is discussion here on the role of the United Nations. In some ways, the previous discussion on social work could also fit in the discussion.
A strong, thoughtful position on genocide versus cultural genocide.
Conclusion
A longer than usual conclusion but understandable with the way the authors have brought in cultural strengths.
References
Multiple relevant references.
Reviewer 2 Report
Comments and Suggestions for Authors
This is a manuscript with a significant strength, concerning the introduction of indigenous voices and perspectives into social science research. This is also an important trend in the contemporary social science, based on ideas of decolonization and confirmation of indigenous perspectives, which needs to be explicitly stated. Relevant references, acknowledging such a methodological perspective, need to be introduced. The historical perspective is relevant, but authors need to be cautious, when drawing the direct comparisons between the past and the current social practices, especially involving very strong statements of discrimination and even ethnocide (e.g. "Ethnocide continues today and can still be seen in policies and practices affecting Indigenous children. Justice work is desperately needed today with the inequitable rates of out-of-home placement of Indigenous children and violations of the Indian Child Welfare Act"). We are not sure what is the scientific contribution of the manuscript, except for providing a review of historical practices. Although this might be OK for the "Perspective" article type, authors are encouraged to position their manuscript within the theoretical perspectives in the extant literature.
Reviewer 3 Report
Comments and Suggestions for Authors
Revisiting One of the Oldest Orphanages, Asylums, and Indigenous Residential Boarding Schools: The Thomas Indian School at Seneca Nation
This manuscript uses the Thomas Indian School, originally and more accurately names the Thomas Asylum for Orphaned and Destitute Indian Children (aka “Salem”) as a lens into settler colonialism in the US and Canada. It then recommends drawing upon Indigenous knowledge and tradition to counter the harms of settler colonialism. The focus of this manuscript is timely and important.
The manuscript is broad and expansive in its focus. While it covers a lot of history and context, it would be stronger if it were more focused. The narrative jumps around and does not build as clearly or logically as it should to bring along readers who are not already sympathetic to the argument. It would be effective to build a clearer historical narrative that shows how initial conquest led to boarding schools, what the boarding schools harms were to Indigenous families using TIS as a case study, and then specifically how Indigenous knowledge can address the systemic of colonization and the boarding schools’ legacy.
A point the authors should focus on is the fact that settler colonialism operates by attacking key institutions of Indigenous societies: the economy, legal, religious, etc. The role of boarding schools was to attack Indigenous families and socialization. For instance, while the state initially forced families to place their children in the schools (I am not sure that happened at TIS), the destruction of Indigenous institutions created conditions where families were forced by abstract forces to place their children in the schools and not directly by the state. This was the case for my family so that two of my uncles, three aunts, and my father were all placed into TIS when the family broke down. That better describes the violence of settler colonialism than abduction that is more consistent with external colonialism. Their prescription therefore can then focus specifically on healing the damage caused to families and how to build Indigenous knowledge. Currently, the manuscript calls for this vaguely but does not provide a map to the goals.
What I recommend is organizing the lit review and history to be more focus on how settler colonialism attacked Indigenous peoples with the Seneca case study vis-à-vis TIS. Then using that case study to make more concrete recommendation on how to draw on Indigenous knowledge to counter the many harms caused by settler colonialism vis-à-vis the boarding school
Project. I think this would make the manuscript more persuasive and give other researcher a clearer a map to engaged research.
I have attached the manuscript since I put a few notes on it.

Reviewer 4 Report
Comments and Suggestions for Authors
I think there are some areas that can be improved:
1. You did not describe the approach you used to organize and prepare the literature review. I am assuming you used a historical perspective. How did you decide what accounts and papers to include? A section describing your approach would strengthen the literature review.
2. Strengths: The literature review is well written provides a comprehensive examination of the historical trauma experienced by Indigenous populations due to Indian boarding schools. The review integrates perspectives from various sources, including academic literature, Indigenous leaders, and historical accounts. The review highlights the ongoing impact of past policies on current social issues.
Weaknesses: The review lacks a critical analysis of the sources cited, such as a discussion on the limitations and/or biases of perspectives presented. While the review touches on the importance of social justice and Indigenous frameworks, consider adding more context, specifically information on the historical background of Indian boarding schools in general and their lasting effects. The manuscript is only 12 pages, so there is room to expand.
Reviewer 5 Report
Comments and Suggestions for Authors
The article offers an important reflection on one of the most delicate and cruel assimilation processes experienced in North America. The historical review of the processes of assimilation, integration and recognition was presented accurately from a historical and social point of view.
The testimonies and documents cited give a good idea of ​​the situation experienced in the Seneca Nation. However, some comparative data with other similar situations in North America is missing. Inserting some data on the number of schools and the types of educational systems in other contexts would help to better understand the situation experienced in the territories considered.
Line 172 - the doctrine of discovery should be explained in greater detail also trying to interpret what the intentions of the European colonists were.
Line 305 - It is very important to cite some data on the ethnocide type, even if these are estimates, but this would help to grasp the dimensions of the phenomenon.
In the conclusions critical considerations should be expressed about a return to ancient practices. It is not certain that this resumption of activity will always be in line with what is necessary (see Mc Arthur 2020 https://www.tandfonline.com/doi/full/10.1080/00131857.2021.1934670)
Round 2
Reviewer 3 Report
Comments and Suggestions for Authors
I am impressed by how the authors responded to my previous review. I like how they reorganized the manuscript so that it builds in a clearer, more logical way. It effectively establishes the harms that boarding schools and TIS in particular inflected on Indigenous communities and how social work as a field was complicit in settler colonialism in order to establish their intervention. I think that this manuscript makes important contributions. I do encourage the authors to do some minor stylistic revisions so that the manuscript reads smoothly and error free throughout the entire manuscript.